# Plant Classification Knowledge and Misconceptions among University Students in Morocco

**Lhoussaine Maskour [1], Anouar Alami [1,* 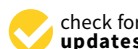], Moncef Zaki [1] and Boujemaa Agorram [1,2]**

[1]   LIRDIST, Interdisciplinary Laboratory of Research in Didactics of Sciences and Technology, Faculty of Sciences Dhar Mahraz, Sidi Mohammed Ben Abdellah University, B.P. 1796, Fès-Atlas 30003, Morocco; lhomaskour@gmail.com (L.M.); zaki.moncef@yahoo.fr (M.Z.); bagorram@gmail.com (B.A.)

[2]   Cadi Ayyad University, Ecole Normale Superieure, EREF, Marrakech 40130, Morocco

*   Correspondence: anouar.alami@usmba.ac.ma; Tel.: +212-661-796-480

**Abstract:** This study aims to assess learning outcomes and identify students' misconceptions in plant classification. We conducted a questionnaire survey with undergraduate and master's students. The qualitative analysis of the students' responses made it possible to shed light on the difficulties of assimilation of many notions and also to identify the different misconceptions constructed during their learning courses about plant organisms. The findings indicate that some students are not motivated to take the course on plant classification. This demotivation is reinforced further by students' perceptions of plant classification, especially that it is not important and not useful for learning other biology specialities. The findings also show that more than half of the students who participated in this study consider plant systematics a difficult subject. We also note that some of the students surveyed seem not to have acquired many concepts of plant biology including concepts related to the biology, reproduction and evolution of plants. Thanks to this, we could see different types of problems in plant classification, which constitute misconceptions hindering learning. Initial training in plant biology does not appear to have a significant effect in modifying students' misconceptions related to plant classification.

**Keywords:** classification; plants; misconceptions; students; teaching

## 1. Introduction

Biodiversity is the basis of human life. Increasingly established in urban areas, human populations largely ignore the extent to which their economic, social and cultural well-being is based on strong and resilient ecosystems characterized by a rich diversity of species. Lack of awareness can lead to practices that overexploit natural resources and damage biodiversity [1,2]. Raising awareness about the role of biodiversity in ensuring environmental sustainability, economic prosperity and social and cultural well-being will contribute to the improvement and effectiveness of sustainable development actions, including ways to develop sustainable development behaviours and sustainable consumption and production at both local and global levels [3,4].

The concern about the loss of biodiversity and the ethical issues related to its sustainable use remain at the heart of education for sustainable development [5–7]. Numerous studies have shown that the most effective lever for preserving biodiversity lies in education and training [8–11]. In this sense, biodiversity education from primary school age, is a major challenge. The child, but also the future citizen, will really understand the need to protect biodiversity only if they develop a rational approach. That is to say if he understands that humans are part of nature and that his activities must take into account the needs of humanity but without depleting the ecosystems that must be preserved for future generations of living beings, whatever they are [8,9]. Biodiversity should be included in

all teaching and learning projects, curricula and teaching materials [12,13]. The learning outcomes sought should cover theoretical understanding, value-building, skills development and the adoption of attitudes conducive to the preservation of biodiversity. Quantification of the effect of education studies attempting to quantify the effect of formal education on biodiversity conservation agree that it has a beneficial effect [14]. For example, one study estimated that between 4 and 21.5 percent less annual area of old growth forest was cut per household for each additional year of education that the household head received, depending on the society being studied [15,16]. The effect however, is non-linear and there is a turning point when the returns from education decrease [16,17]. The positive influence of education also depends on the type of conservation being carried out. For example, Gotmark, in a study in Sweden, shows that education contributes to the conservation of mature trees but not to the planting of saplings [18].

Teachers should prepare students to face the real problems they will encounter on a regular basis in their efforts to sustainably manage the biosphere and integrate biodiversity conservation with other societal goals [19–21].

In addition, to preserve plant and animal species, we must first know them, know their biology and know how to identify and classify them.

For centuries, biologists have worked on classifying organisms in a way that would help to clarify the relationships between species over time and in different environments. In trying to delimit the order of living beings on earth, they faced a complex mission. Some estimate that 5 to 40 million living species inhabit Earth's lands and waters. So far, scientists have managed to classify and name only about 1.6 million species, including about 300.000 plants [22].

Plant classification has undergone very important changes throughout history. In fact, at first, the classification of plants was first based on their utility (food property, therapeutic property) and then on their morphology (organization and arrangement of the different floral parts, reproductive organs etc.). They are now based on the search for common ancestors (phylogeny or cladistics) thanks to effective methods of analysis (molecular biology, sequencing of genes). Plant taxonomy teaching is an important part of botany in universities and is the focus of all the reforms of this branch of education [23,24]. In order to eradicate the existing problems in plant taxonomy teaching, some reforming measures have been put forward. Some of them focus on innovation in this field, the teaching content, the student teacher's rapport or the reduction of teaching time [23,24]. Others put emphasis on the adequacy of theoretical backgrounds and practical activities [17,25].

However, literature shows that many student misunderstandings about plants and their classification continue to exist [26–31]. Thus, research has shown that many children aged 10 to 12 do not perceive plants as living beings. Many children from this category think that they come from shops or that wild flowers grow in the fields only because humans plant them or because they are parts of land [32–34]. Secondary students (13–15 years) even noted that they do not consider plants to be living things [35]. Other studies have shown that elementary and secondary students have problems in classifying and understanding the diversity of living organisms [32,36]. To study misconceptions about classification, most of these studies were conducted among students in elementary and high school. A common finding of these studies shows that students' conceptions about plants are unscientific. The main misconceptions reported in these studies are that students establish a direct link between some seedless plants and invasive plants, vascular plants and non-vascular plants, gymnosperm plants and angiosperm ones [35].

Students tend to classify plants according to recognizable characteristics (green colour, growing in the soil etc.) and different parts of the plant (stem, leaves, flowers etc.). For example, about half of the students in one study classified a fungus as a plant because its stem resembles the stem of a plant [32,37,38]. Students may also not consider trees as plants. However, this may be due to students' limited skills in classifying rather than misunderstanding plants. Other researchers have found that when classifying animals, elementary students tend to use mutually exclusive groups rather than subsets of a larger group. This can also apply to plants [38].

As far as university students are concerned, there are few studies about them [39,40]. Yangin, who worked on future teachers, found that almost all participants in his study confounded fungi with plants and linked some gymnosperm plants to angiosperm ones [37]. Misconceptions could be acquired from students' own experiences in life before they enter school, through media, textbooks or due to the bad quality of teaching [37].

If science education is to instil in students different biological concepts, such as living things, plants, and animals, then it is essential to identify the erroneous or alternative conceptions of these concepts and to plan teaching activities which reinforce or question their previous conceptions [41,42]. This approach, based on the identification of students' alternative conceptions and their use in teaching activities, has been tested in science teaching, thanks in particular to several studies [42,43]. Studying students' conceptions about plants and classifying them provides an assessment of the state of knowledge and then remedies and adjustments can be applied to teaching methods [37].

The present study aimed at evaluating learning of plant classification in Moroccan universities by analysing the achievements of students and identifying their misconceptions about plants and their classification. Thus, the research questions examined in this study are as follows:

1.     What are the students' perceptions of the interest and importance of plant classification?
2.     What are students' knowledge about plants and their classification?
3.     Are students' conceptions about plant classification consistent with science?

## 2. Materials and Methods

### 2.1. Contents Related to Plant Classification in the Science Program at the University (Life Sciences Section)

Among the 24 modules that make up the Life Sciences section, some are related to plant classification. These modules are taught in lectures, tutorial and practical works. Table 1 shows the distribution and content of these modules during the three years of the Bachelor studies.

**Table 1.** Distribution of modules related to plants and their classification during studies of Bachelor's degree.

| | **Modules and Content** |
|---|---|
| High School Diploma +1 year | Cell Biology module: Acquisition of basic notions about the structure and functions of plant and animal cells components (28 h of lectures and 15 h for tutorial and practical works). |
| | Plant Biology module: The basics of lower and higher plants. -General characteristics of Thallophytes. General characteristics of Cormophytes., -Algae -Organization of Phanerogams (vegetative and reproductive system) -Introduction to the histology of Angiosperms. (24 h of lectures, 6 h of Tutorial works and 15 h of practical works). |
| High School Diploma +2 years | Plant biology and physiology" module: It focuses in depth on various fundamental aspects of plants and their relationship to the environment. (12 h for tutorial works and 30 h for practical works). |
| High School Diploma +3 years | In semester 5, which is the common core for the BCSs of all specialties: Biology of Organisms and Ecosystems (plant kingdom): theoretical and practical basics of botanical systems. It allows students to become acquainted with the flora especially with groups of regional importance. (26 h of tutorial work and 20 h of practical works). Biosystematics Module: History and principles of the classification in the plant kingdom and the classification and evolution of large plant groups. The practical work deals vegetative, floral and fruiting organs of species belonging to the main families in the region. |
| | Plant Ecology module: (20 h of lectures and 4 h of tutorial works and 6 h of practical works. |

All in all, we can conclude that the surveyed students study at least three modules about plants, including at least one devoted to systematics.

*2.2. Research Design*

In order to study knowledge and misconceptions related to plant classification among university biology students, we used a survey research design. These research methods are suitable for this study because types of misconceptions and perceptions are the key variables of this study.

*2.3. Population and Sample*

2.3.1. Population

The study population is made up of students from some Moroccan universities: (Faculty of Sciences Ben M'Sik-Casablanca (254 students), Faculty of Sciences Ibn Zohr-Agadir (190 students), Faculty of Sciences Fes (201 students) and Ecole Normale Supérieure Marrakech (92 students).

2.3.2. Sample

Our sample consists of 737 students. The gender ratio of study participants was 63.2% female to 36.8% male. The average age of students is 22.05 years (± 2.10). The sample covered the following university levels: High School Diploma + 2 years (277 students), High School Diploma + 3 years (Bachelor of science degree) (266 students) and 194 studying for the master's degree. Students are categorized by specialty of studies as follows (Figure 1).

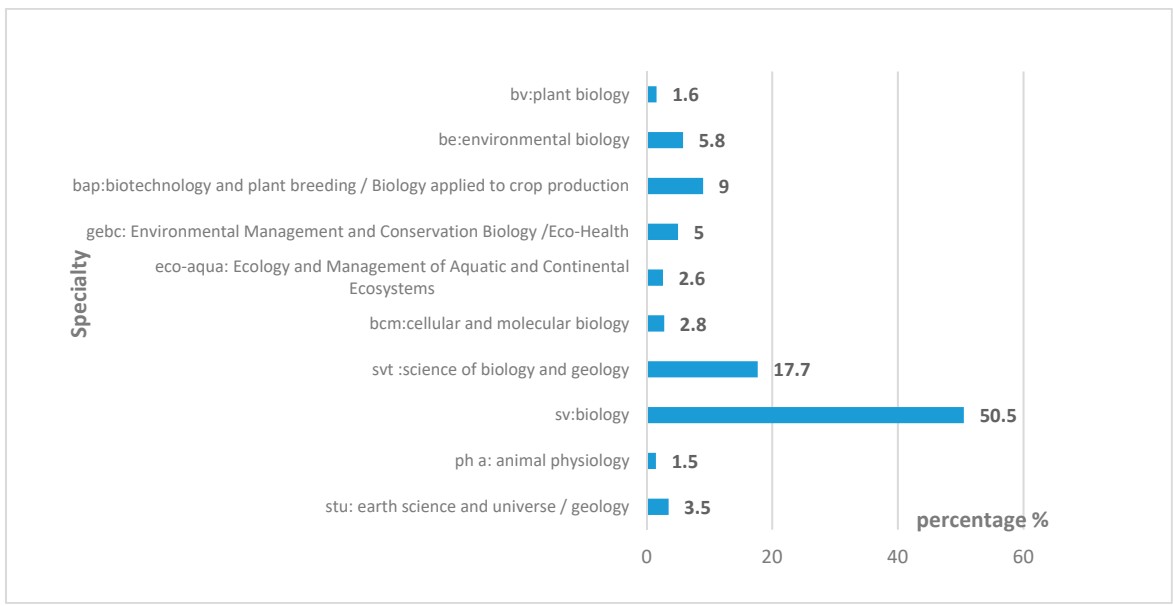

**Figure 1.** Distribution of students by specialty of study.

All the students of the sample had completed two years of study in the Life Sciences section. So, they had studied the following modules: Cell Biology, Plant Biology1, Plant Biology and Physiology, Biology of organisms and Ecosystems and Biosystematics.

*2.4. Ethical Approval and Data Collection*

The authorization to survey students had been previously obtained from the institutions concerned. The survey was introduced by explaining the purpose and goals of the study. Participants were asked to participate and were informed of the guarantee of the anonymity. Participants were also informed that they can refuse to participate in the survey and that those who give their consent

are invited to complete the questionnaire. The survey was administered on paper. This was more time-consuming than electronic surveys but this method was chosen to increase response rates.

### 2.5. Survey Instrument

The survey instrument (see Appendix A) was developed using student misconceptions identified in the literature [29,35,37–39,44–46]. The knowledge questions were graded taking into account the orientation of the content of the curriculum of plant biology at university in Morocco.

The survey instrument was pre-tested on a sample of 30 students from Semlalia faculty of Sciences (these students have not participated in the final survey). Items are written in French. The best formulation was selected during a consensus meeting with content knowledge and French language experts.

The validity of the instruments was established by subjecting the instruments to the expert judgment of experts in the field of plant biology and didactics of biology in the departments of Biology and Sciences didactics in the Ecole Normale Superieure.

In order to ensure reliability of the instruments, a reliability coefficient was computed using the Cronbach Alpha method of reliability to establish internal consistency of the instrument. In this case, the level of significance in which the instruments were adjudged reliable was at 0.74.

Nine hundred copies of the questionnaire were distributed according to the available means and access conditions (return rate was 81.9%). Participants completed the questions individually. The identity of the participants remained anonymous.

The questionnaire has 135 questions. The survey questions were classified into four sections to facilitate understanding of the questions and to avoid any possible ambiguity (in this article, we treat a part of the questionnaire (see Appendix A for the full questionnaire). The questionnaire allowed the collection of information on the following elements:

- Students background information about their age, sex, level of education.
- Student perceptions about the difficulty and importance of plant classification learning
- Students opinions about plant classification teaching and learning methods
- Student knowledge and conceptions about plant and their classification.

## 3. Results

### 3.1. Marks Obtained in Plant Classification and Plant Biology

Generally, the marks obtained by the students in plant biology course are better than those obtained in plant classification courses. In plant classification courses, 1/3 of students report having lower than average marks (10/20 that correspond to "grade B" or "satisfactory" in English school marks system) and grades of 2/3 of students are between 10 and 15/20 (grade B to A⁻ in the English school marks system). In plant biology courses, only 1/6 of students scored below average while about 5/6 of them scored above average.

### 3.2. Students Perceptions of Level and Causes of Difficulty in Plant Classification Learning

More than half of the students (53%) consider plant systematics a difficult subject and 1/3 (36%) consider it to be moderately difficult. The majority of students (59%) state that the difficulty of learning plant systematics is due to Latin nomenclature; 1/3 of students say that the nature of this subject is complex and difficult and more than 1/5 of students state that this difficulty is due to teaching (methods, program, number of teaching hours, etc.).

### 3.3. Perceptions of Students about Plant Classification Importance

Three out of four students said that plant systematics is important (74.2%) while one out of four considered it to be a little or not important discipline (25.8%).

We found that 23% of students state that the use of taxonomic knowledge and knowledge about plant systematics and flora in other subjects of biology is low, while 41% of them consider that this use is moderately important and only 36% think this use is very important.

To implement practical actions to improve species classification skills, students must first be aware of their importance and be motivated to develop them. However, a significant proportion of students think that plant classification knowledge is not important for other disciplines of biology. Therefore, any measures to improve plant identification and classification skills must take into account the reasons why some students are willing to acquire plant identification skills and why some think they are unimportant in biology. This is a line of research that needs to be developed further using other more qualitative research techniques.

### 3.4. Perceptions about Learning Methods

Figure 2 present students' responses relatives to their perceptions about plant classification learning and their interest to it.

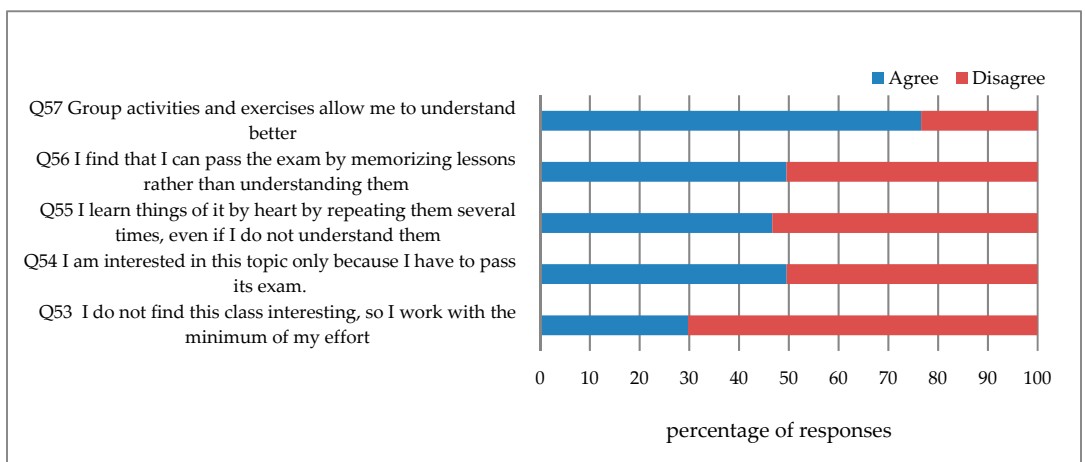

**Figure 2.** Students' perceptions about plant classification learning.

We found that almost two out of three students find this subject interesting (70%). Half the students state that they learn by memorization and rote learning rather than by comprehension.

### 3.5. Perceptions of Students Regarding the Definition of Plant Classification

In Figure 3, questions Q89 and Q90 define classification in the phylogenetic context.

Nearly half of students (44.2%) consider that plant taxonomy determines "which plants are close to each other", whereas 37.5% of students' conceptions of plant taxonomy were related to an evolutionary problem. The majority of past species remain intact and fossilized. Phylogenetic classification takes them into account, comparing them with current species. Hence, it can link relatives by defining their attributes and using the relevant traits. Almost two out of five students (39.3%) think that the classification of living beings does not consider current species in a similar way to fossil species. Students seem to think that current species are more important in taxonomy than extinct or fossil species because the characteristics needed for identification and classification (morphological, anatomical, biochemical or genetic characters) are easier to use in current species than in fossilized ones (especially for DNA-based identification). Similarly, 49.9% of students think that plant taxonomy's main objective is to classify plants and sort them, thus expressing functionalist conceptions without referring to evolutionary aspects.

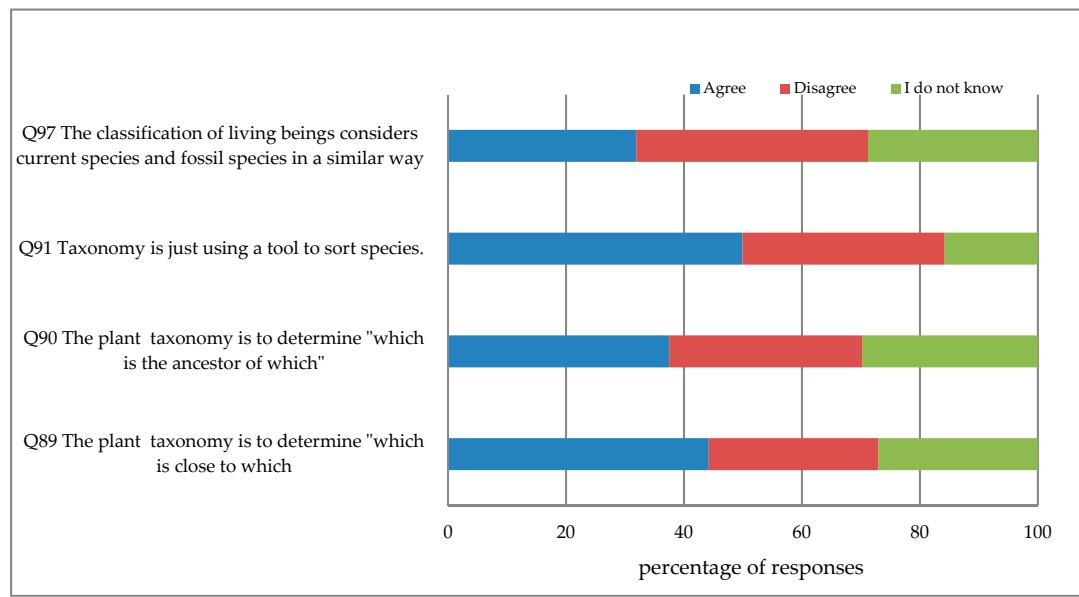

**Figure 3.** Students' definitions of plant classification.

### 3.6. Perceptions of Utility of Plants' Classification

In Figure 4, we notice that the majority of students (69.52%) are motivated to learn how the plant species studied are useful and 60.14% of students consider that a species of medical use is more important than other species without medical use. Moreover, a large number of students consider that knowledge of plant species serves only the knowledge of their food and medical uses (40.16%) or their ecological importance (33.47%).

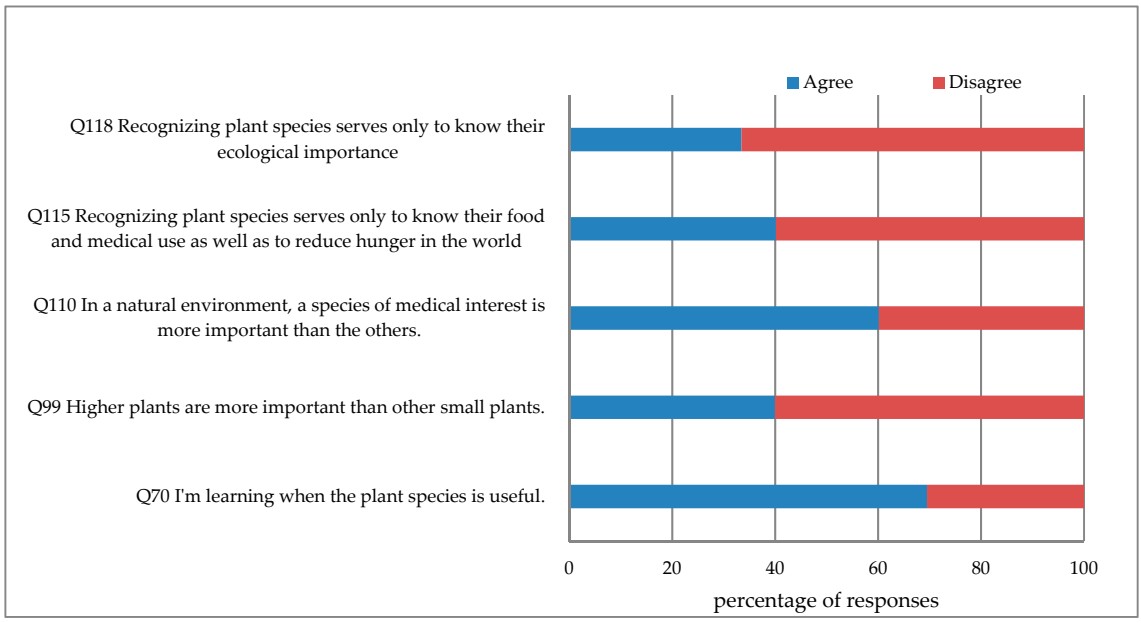

**Figure 4.** Students' perceptions of importance of plants and their classification.

### 3.7. Perceptions of Students Related to Plant Classification Objectives

The majority of students surveyed are aware of the relationship between plant classification and understanding of biodiversity (79.22%) and its preservation (77.64%) (see Figure 5). Also, they understand the importance of teaching classification even though there is an increasing decline in many species (72%). However, only 64.30% of students surveyed understand the importance of plant

classification for understanding plant unity. Moreover, almost one student out of three (36.28%) thinks that the main objective of plant classification is to make herbaria and enrich gardens.

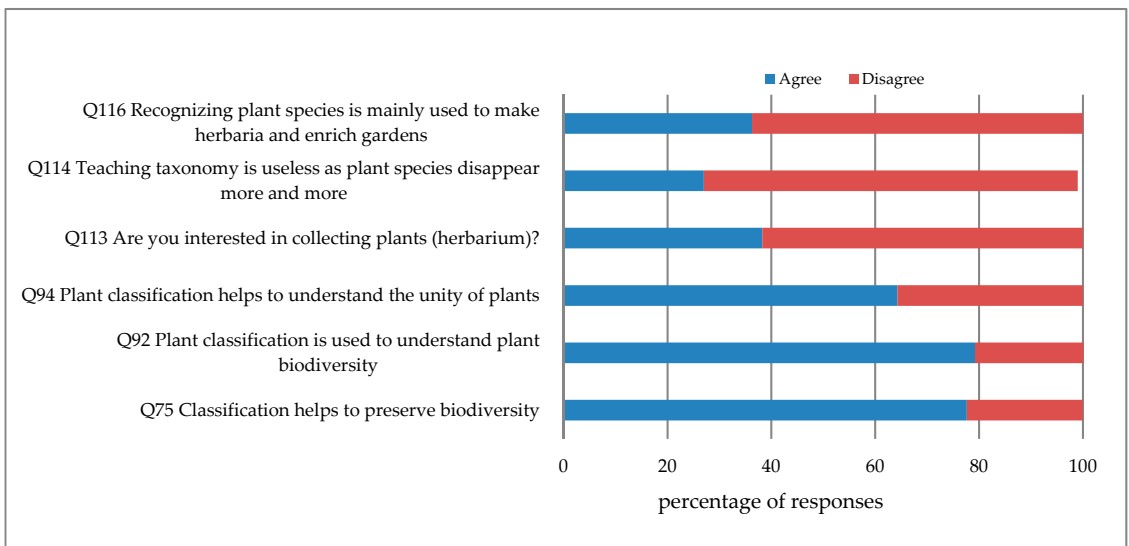

**Figure 5.** Students' perceptions of interest of plant classification.

### 3.8. Students' Knowledge about Reproduction

The results are presented in Figure 6 below:

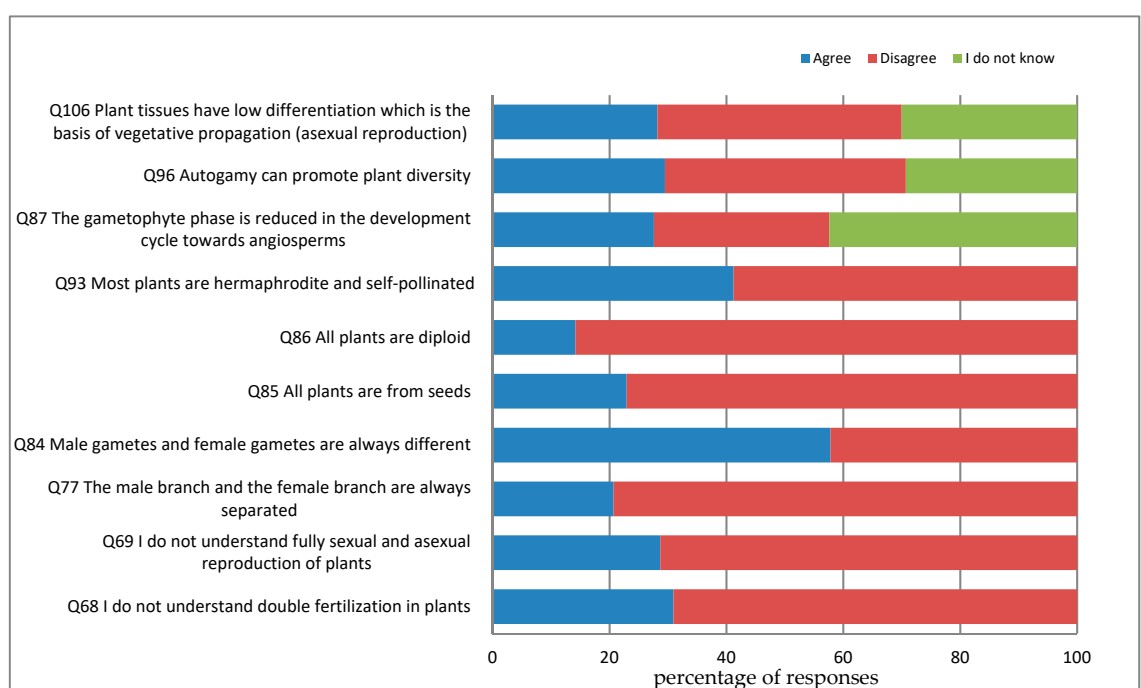

**Figure 6.** Students' knowledge about plants (Reproduction).

Despite the fact that double fertilization is part of the studied concepts in the unit of sexual reproduction in plants (at the level of secondary education and university), almost one out of three students declare that they are unable to understand this concept (30.94%). The same goes for the notions of the sexual and asexual reproduction of plants (28.70%). We also noticed that about two out of five students think that most plants are self-pollinating (41.24%), one out of five students declare that most plants are monoecious (20.71%) or that all plants are diploid (14.19%).

We also noticed that more than three out of ten students could not answer questions Q87, Q96 and Q106 (42.4% for question Q87, 29.3% for Q96 and 30% for question Q106). These questions concern notions relating to plant reproduction (place of the gametophyte and sporophyte phases in the life cycle of angiosperms, autogamy reduces diversity in plants) or to histology that are poorly understood by the students. About 30% of the students surveyed think that autogamy favours plant diversity and that the gametophyte phase is not reduced in the development cycle towards angiosperms (30%) or that plant tissues have a low number of differences (41.8%). These notions are nevertheless taught in courses of plant biology and plant classification .

### 3.9. Students' Perceptions about Plant Evolution (Phylogeny)

The results are presented in Figure 7 below:

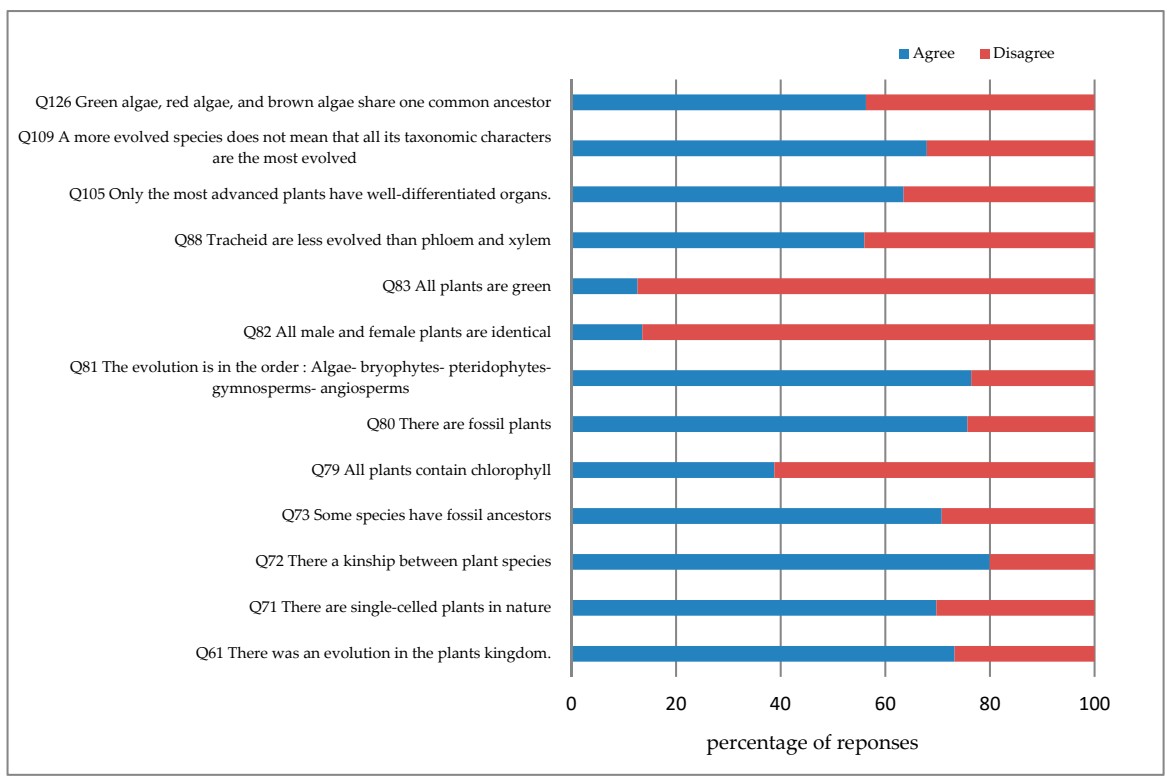

**Figure 7.** Students' knowledge about plants (Biology).

It can be seen that a few students were not able to correctly answer questions related to notions they had already studied. Thus, 38.8% of students surveyed believe that all plants contain chlorophyll or that tracheids are less advanced than phloem and xylem (43.99%).

Similarly, 23.58% of students do not know the order of evolution within the plant kingdom; others seem to refute the existence of evolution in the plant kingdom (26.74% of students).

### 3.10. Some Misconceptions İdentified Among Students

The main misconceptions identified among students are presented in Figure 8 below:

Almost four out of ten students (39.9%) could not answer questions related to notions they had already studied.

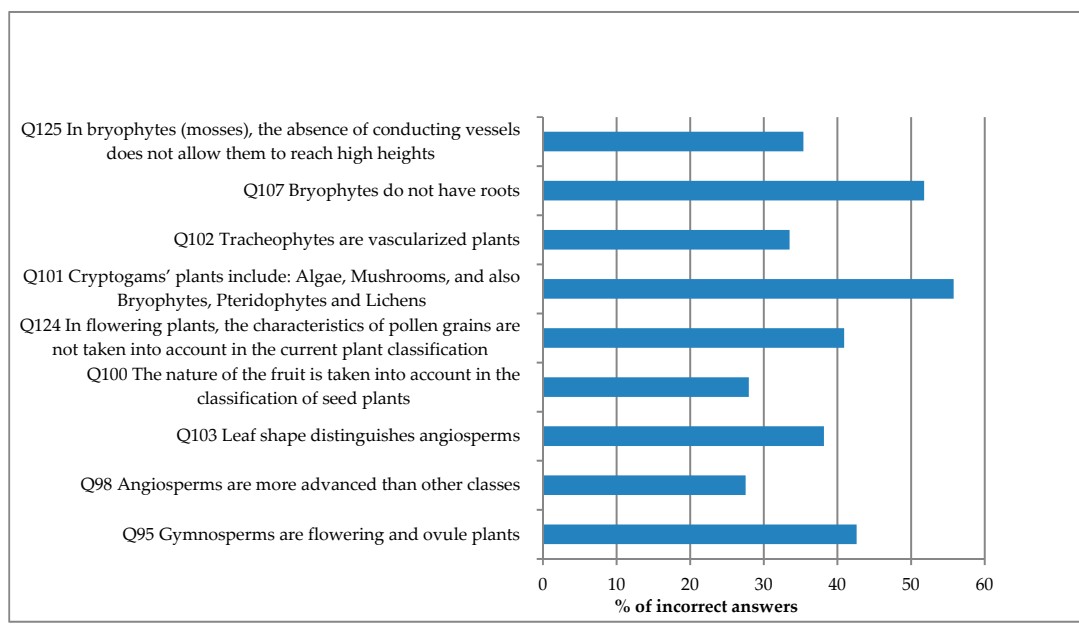

**Figure 8.** Some misconceptions identified among students.

## 4. Discussion

More than half of the students who participated in this study consider plant systematics a difficult subject. Among the causes of this difficulty mentioned by the students we find Latin nomenclature used to name the species or the nature of the discipline itself because it involves other disciplines of biology (plant biology, cell biology, histology etc.). Faced with the task of classifying plants, students had to mobilize knowledge they had studied previously during their school curriculum (morphological and anatomical characters to differentiate taxonomic entities, floral diagram, floral formula, reproduction in plants, etc.). This constitutes an obstacle difficult to overcome because of the forgetting or non-assimilation of this knowledge. Other students blame the methods used in teaching taxonomy because they favour memorization at the expense of reflection and comprehension. Traditionally, plant classification courses focus on transmitting knowledge to students through identifying and naming plant species. This acquisition of knowledge by memorizing facts, naming, describing and identifying constitutes basic skills according to Bloom's taxonomy, while knowledge and comprehension are central skills to learn in plant classification coursework.

Not diversifying the contexts of teaching (practical work to study some plants, scarcity or absence of study outings to fields) also weakens the motivation of students to study this subject. Moreover, numerous studies have shown that, in the school context, to guarantee effective learning and academic success students must be well motivated [47,48]. This demotivation is reinforced further by students' perceptions of plant classification, especially that it is not important and not useful for learning other disciplines of biology. Demotivation problems are usually associated with learning difficulties. Students feel incompetent in the study of plant classification. This feeling of incompetence will negatively impact students' motivational dynamics and academic success [47].

The students surveyed herein define plant classification in different ways. Almost half of them assert that plant taxonomy's main objective is to determine "which species are close to which." This is classification with a phylogenetic aspect. A phylogenetic classification responds to a historical problem and is part of the evolutionary paradigm [49]. In the context of phylogenetic thought, the problem is: Which taxon is the closest to which in terms of evolutionary history? A phylogenetic group is monophyletic and gathers related species having an exclusive common ancestor. It is this principle that underpins the current conception of phylogenetic classification and which is translated schematically by representations in the form of a phylogenetic tree. Therefore, grouping organisms sharing the same evolutionary origin does not require taking into account all similarities but those inherited from a

common ancestor [48]. Some students also think that plant taxonomy is limited to classifying plants. Thus, this view reduces the taxonomy to the use of a tool to sort species ignoring its other functions of classifying, describing and naming the species. These functions complete the definition of the classification. These confusions concerning the definition of classification must challenge us in the effectiveness of the current teaching of the classification of plants and the living beings in general. This way of teaching must be renewed by making profound changes [50,51].

Some researchers propose that plant classification should transition from a traditional passive lecture format to an active-learning curriculum, with focus on student-centred pedagogies such as collaborative in-class group work and discussion to increase student engagement [52,53]. Some of proposed activities include student-led field presentations of plants and individual research, while incorporating several platforms for introduction and review including discussions on problem solving and games and online tools and mobile applications [52,53]. Also, doing field trips makes students more motivated to learn botany and plant classification activities. The field environment is essential for exposing students to live plants in their habitat. Thereby they can better understand phenology and diversity within a single species, morphological variation through space and time and developmental stages of the plant [23,52,53].

Nearly three quarters of students say they learn more when the plant species studied are useful and especially when it comes to a species with a medical use. Some of students surveyed think that recognizing plant species only serves to know their food and medical uses for reducing hunger in the world. This is a utilitarian vision that results from a human-centred ethic [50]. This anthropocentric vision states that all natural resources are at the unconditioned disposal of humanity. It is therefore important to know how to increase students' motivation to learn techniques of identification of plants that have no practical use or cultural significance. It can also be noted that textbooks in Moroccan secondary schools often include higher plant species (tree, shrub); thus, students become familiar with species with large sizes. It is also noticed that despite their university education, there are students who believe that higher plants are more important than other lower plants. These students neglect the ecological and utilitarian importance of lower plants, reflecting a macro-centric view that focuses only on large-scale terrestrial plants.

In addition, the majority of students surveyed are aware of the relationship between plant classification and the understanding of biodiversity and its preservation. Indeed, to monitor the evolution of this biodiversity and safeguard it, the identification of all species is essential. However, almost two-fifths of students reduce the objective of plant classification to herbaria and enriching gardens.

In functional classification, the characteristics associated with reproduction cause difficulties for the students. Concepts such as double fertilization or sexual and asexual reproduction are poorly understood by students, despite the fact that these notions are studied in both secondary and university education.

In relation to plant reproduction, we can identify several misconceptions such as: the majority of plants are allogamous, monoecious and diploid; autogamy favours the diversity of plants; male branches and female ones are always separated; the gametophyte phase is not reduced in the development cycle towards angiosperms; there is a weak difference between plant tissues; all plants come from seeds; all plants are green. These results have been found in other studies on students' conceptions about plants [32,37].

In another setting, over a quarter of students seem to not believe in plant evolution. Evolution is seen as an important theme that brings to the school a broader perspective of natural phenomena and the nature of science. Research has shown that the teaching of theories of evolution has not had a positive impact on the understanding and acceptance of evolution in different countries of the world [54–56]. Students do not understand the theories of evolution correctly. They believe that evolution has been primarily related to the human species. These misconceptions are not only related to teaching methods and cognitive abilities of students but also to the evolution of factual knowledge

and epistemological barriers that could be the result of a process of social reconceptualization of knowledge offered to students [54,55].

As for the structures and conductive tissues, more than two out of five of the students questioned do not consider that the tracheids are less evolved than the phloem and xylem which characterize gymnosperms and angiosperms. Tracheids are low-skilled conductive structures present particularly in Pteridophytes. The plants are divided into vascular and non-vascular groups. Vascular plants (pines, ferns, corn, etc.) have tubes called xylem and phloem to carry water and food throughout the plant. In contrast, non-vascular plants (e.g., mosses) do not have these tubes and transfer food and water from one cell to another [29,35].

It seems that other notions of botany are not understood by students such as the fruit and the flower and their use as important characteristics in the classification of plants. Vascular plants are divided into three main groups: angiosperms, gymnosperms and ferns. Angiosperms produce fruits and flowers, gymnosperms have parts of seeds in cones (pines) and ferns are the third most important type of vascular plants and they have no flowers, no fruit and no seeds. Angiosperms are flowering plants. Gymnosperms include non-flowering primitive plants such as conifers. Ferns produce spores from which new plants grow [27,35,38].

The flower and the fruit are thus characteristics that make it possible to gather species in larger units. The role of the systematics scientist is to determine the relevant characteristics to differentiate plants and to classify them into homogeneous groups in a perspective that explains the biodiversity on earth and the kinship of all living beings in a phylogenetic context [57–60]. This is not an easy task because it is difficult to find homologous traits, especially since these traits often change with plant development [45].

## 5. Conclusions

Currently, plant classification is increasingly devalued at the expense of other disciplines of biology. The same goes for teaching this discipline. Students are less and less motivated for classification and there is unfortunately a decline of researchers interested in the few laboratories still working on systematics. This resulted in a decline of specialists in this field. This situation is caused by many factors. Teaching the classification of living beings must be examined and improved.

In this study, many misconceptions which could be barriers for students learning plant classification were identified. Students have different ideas about plant classification and its objectives, including distinguishing and clarifying the complex relationship between definition and classification activities. Theses misconceptions are identical to those mentioned in various previous studies [32,37,38,46,58]. The plurality of classifications is also a major difficulty to overcome; students often confuse the different classifications (utilitarian, functional, phylogenetic) [59,60]. A scientific classification should be constructed and it should be set up within a context of explaining and showing the diversity and kinship of all living beings and which also takes into account their history.

Educational program designers must also integrate diversified species, covering all higher and lower plant groups and taking into account their ecological and environmental importance. Moreover, university education should allow students to learn to distinguish the scientific repertoire from the social and religious one; therefore, training in plant taxonomy must be rectified at all university levels to allow students to learn an entirely evolutionary thought developed transversally to construct plant phylogenetic classifications.

**Author Contributions:** Conceptualization, L.M., A.A. and B.A.; methodology, L.M. and B.A.; formal analysis, L.M., A.A., B.A. and M.Z.; investigation, L.M. and B.A.; writing—original draft preparation, L.M.; Writing, review & editing, L.M., A.A., B.A. and M.Z.; supervision, A.A. and B.A.; project administration, A.A. and M.Z.; funding acquisition, L.M., A.A., B.A. and M.Z.

**Funding:** This research received no external funding.

**Acknowledgments:** We thank Ahmed Ouhammou from Cadi Ayyad University for scientific and technical support he provides us.

**Conflicts of Interest:** The authors declare no conflict of interest.

**Appendix A**

| Questionnaire |
|---|
| As part of the preparation of a doctorate on The teaching of plant classification at the university, we undertake this survey, of which the following questionnaire is part. We request your cooperation by asking you to fill out the questionnaire which is anonymous |

| | < 5/20 | between 5 and 10/20 | between 10 and 15/20 | > 15/20 |
|---|---|---|---|---|
| - **Gender**: *Man* *Women* <br> -**Spéciality**: . . . . . . . . . . . . . . . . . . . . . . . . . . . . . . . . . . . . . . . . . . . . . . . . <br> - **University level:** . . . . . . . . . . . . . . . . . . . . . . . . . . . . . . . . . . . . . . . . . . . . <br> - **if you have failed a class, how many times have you failed:** . . . . . . . . . . . . | | | | |
| Q1- Marks obtained in floristry/plant Systématic | | | | |
| Q2- Marks obtained in plant biology | | | | |

Q3-What do you think about the importance of plant systematics?

*Very important.* *mportant.* *Little important* *Not important*

Q4-How difficult is plant systematics?

*very difficult* *difficult* *moderately difficult* *easy*

Q5-The difficulty of teaching plant systematics is due to:

*Teaching methods* *subject itself* *Latin nomenclature*

☐ *other:* ……………………………………………………………………………………………………………

Q6-According to you, the need for knowledge of plant systematics and flora in other subjects of biology is:

*Very High* *High* *More or less high* *low* *Very low*

| For each of the proposals below, choose the appropriate answer: | | Never | little | sometimes | Often |
|---|---|---|---|---|---|
| | **Regarding the use of resources and communication tools** | Never | little | sometimes | Often |
| Q7 | The digital resources made available by the teacher integrate only schemas and images. | | | | |
| Q8 | The digital resources provided by the teacher integrate real photos of plants and videos | | | | |
| Q9 | The teacher uses various tools in class: blackboard, slides, video, . . . | | | | |
| Q10 | the French language used in the course poses problems for you? | | | | |
| Q11 | The technical and Latin words used in the course poses problems for you? | | | | |
| | **Regarding teaching methods:** | | | | |
| Q12 | The teacher presents the course as a presentation and distributing the boards. | | | | |
| Q13 | During the class, you have trouble taking notes? | | | | |
| Q14 | The teacher shows you tips for making learning easier. | | | | |
| Q15 | The teacher encourages group work during tutorials and practical work | | | | |
| Q16 | The teaching of floristics is closely linked to daily and professional life. | | | | |
| Q17 | The explanations provided by the teacher, in the lecture course, tutorial and practical work are sufficient | | | | |

| | | In term of motivation: | no | rather no | Rather yes | yes |
|---|---|---|---|---|---|---|
| Q18 | | I am very motivated in the course of plant systematics | | | | |
| Q19 | | The tasks that are proposed to me are very interesting | | | | |
| Q20 | | I ask questions in the course | | | | |
| Q21 | | The course makes me want to learn | | | | |
| Q22 | | I want to do more research in plant systematics | | | | |
| | | In terms of Activities and Interactions: | | | | |
| Q23 | | I learn more in quantity | | | | |
| Q24 | | The teacher encourages you to do presentations | | | | |
| Q25 | | There is more often work to be done in groups (binomial or more). | | | | |
| Q26 | | I have more interactions with the teacher during tutorial and practical work | | | | |
| Q27 | | There is interaction between the students during tutorial and practical work | | | | |
| | | In terms of Assessment: | | | | |
| Q28 | | The exam covers exercises far removed from those studied in classroom and TD. | | | | |
| Q29 | | The exam is in the form of multiple choice questions | | | | |
| Q30 | | The exam questions are more about knowledge | | | | |
| Q31 | | I have more good grades in practical works | | | | |

| | Questions about hourly volumes of the Floristic module | excessive | Adequate | insufficient | very insufficient |
|---|---|---|---|---|---|
| Q32 | The number of class hours is in your opinion | | | | |
| Q33 | The number of hours you spend on home to revise the floristic course is | | | | |

| | Tutorial work (called TD) | Strongly disagree | Disagree | Fairly agree | agree | I don't know |
|---|---|---|---|---|---|---|
| Q34 | The coordination between the lecture courses and tutorial work is well done | | | | | |
| Q35 | During tutorial work, the teacher explains better than in the lecture courses | | | | | |
| Q36 | The tutorial work allows students to learn concepts not treated ed in the lecture courses | | | | | |
| Q37 | According to you the number of hours of tutorial work is excessive | | | | | |

| | Practical works (called TP) | Strongly disagree | disagree | Fairly agree | agree | I don't know |
|---|---|---|---|---|---|---|
| Q38 | The objectives of each TP are announced at the beginning of the session. | | | | | |
| Q39 | According to you the number of TP is sufficient | | | | | |
| Q40 | The time allocated for each TP is sufficient | | | | | |
| Q41 | Students actively participate in the TP | | | | | |
| Q42 | The TP facilitate the understanding of lecture course | | | | | |
| Q43 | The TP completes the lecture course | | | | | |
| Q44 | The controls at the beginning of each TP session oblige me to prepare it in advance | | | | | |

| | Outdoor activity in plant ecology: practical activities | Strongly disagree | disagree | Fairly agree | agree | I don't know |
|---|---|---|---|---|---|---|
| Q45 | Students are informed of the goals of each outdoor activity | | | | | |
| Q46 | The concepts treated during outdoor activities in plant ecology have nothing to do with the teaching of floristics | | | | | |
| Q47 | the number of activities outdoors is sufficient | | | | | |
| Q48 | The number of outdoor activities is sufficient. | | | | | |
| Q49 | The outdoor activities of Plant Ecology makes it possible to apply the notions treated in the lecture course | | | | | |
| Q50 | The outdoor activities allow an important implication of the students in learning | | | | | |
| Q51 | I am satisfied with what I learn in plant systematics | | | | | |
| Q52 | I think that all subjects of plant systematics can be interesting if I understood them. | | | | | |

| | Indicate your agreement with each of the following proposals | Absolutely Agree | Agree | Little agree | Disagree |
|---|---|---|---|---|---|
| Q53 | I do not find this class interesting, so I work with the minimum of my effort | | | | |
| Q54 | I am interested in this topic only because I have to pass its exam. | | | | |
| Q55 | I learn things of it by heart by repeating them several times, even if I do not understand them | | | | |
| Q56 | I find that I can pass the exam by memorizing lessons rather than understanding them | | | | |
| Q57 | Group activities and exercises allow me to understand better | | | | |
| Q58 | The program this module is overloaded | | | | |
| Q59 | I learned bad because the course is delivered too quickly | | | | |
| Q60 | I learn the names of plants easily | | | | |
| Q61 | There was evolution in the plant kingdom. | | | | |
| Q62 | The order of the plant systematic teaching is identical to that of the evolution of plant groups. | | | | |
| Q63 | Generally the exam in this module is easy. | | | | |
| Q64 | In the exam there were questions about the interest of the studied species | | | | |
| Q65 | After the final exam, I feel able to sort the plant species | | | | |
| Q66 | After the final exam, I find myself able to describe different plant species | | | | |
| Q67 | In lab and ecology outdoor activities, I find myself able to identify the plant species | | | | |
| Q68 | I do not understand double fertilization in plants | | | | |
| Q69 | I do not understand fully sexual and asexual reproduction of plants | | | | |
| Q70 | I learn better when the studied plant is useful. | | | | |
| Q71 | There are single-celled plants in nature | | | | |
| Q72 | There a kinship between plant species | | | | |
| Q73 | Some species have fossil ancestors | | | | |
| Q74 | A taxon includes individuals who resemble each other | | | | |
| Q75 | Classification helps to preserve biodiversity | | | | |

| Indicate your agreement with each of the following proposals | | Absolutely Agree | Agree | Little agree | Disagree |
|---|---|---|---|---|---|
| Q76 | A species gathers individuals of identical morphology that can reproduce with each other? | | | | |
| Q77 | The male branch and the female branch are always separated | | | | |
| Q78 | Even if two species are identical, they can be different genetically. | | | | |
| Q79 | All plants contain chlorophyll | | | | |
| Q80 | There are fossil plants | | | | |
| Q81 | The evolution is in the order Algae-bryophytes-pteridophytes-gymnosperms-angiosperms | | | | |
| Q82 | All male and female plants are identical | | | | |
| Q83 | All plants are green | | | | |
| Q84 | Male gametes and female gametes are always different | | | | |
| Q85 | All plants are from seeds | | | | |
| Q86 | All plants are diploid | | | | |
| Q87 | The gametophyte phase is reduced in the development cycle towards angiosperms | | | | |
| Q88 | Tracheid are less evolved than phloem and xylem | | | | |
| Q89 | Plant taxonomy involves determining "which plants are close to eachother" | | | | |
| Q90 | Vegetable taxonomy involves determining "which is the ancestor of which" | | | | |
| Q91 | Taxonomy is just using a tool to sort species. | | | | |
| Q92 | Plant classification is used to understand plant biodiversity | | | | |
| Q93 | Most plants are hermaphrodite and self-pollinated | | | | |
| Q94 | Plant classification helps with understanding the unity of plants | | | | |
| Q95 | Gymnosperms are flowering and ovule plants | | | | |
| Q96 | Autogamy can promote plant diversity | | | | |
| Q97 | The classification of living beings considers current species and fossil species similarly | | | | |
| Q98 | Angiosperms are more advanced than other classes | | | | |
| Q99 | Higher plants are more important than other small plants. | | | | |
| Q100 | The nature of the fruit is taken into account in the classification of seed plants | | | | |
| Q101 | Cryptogams' plants include: Algae, Mushrooms and also Bryophytes, Pteridophytes and Lichens | | | | |
| Q102 | Tracheophytes are vascularized plants | | | | |
| Q103 | Leaf shape distinguishes angiosperms | | | | |
| Q104 | To identify a species, just use a key determination based on observable characters | | | | |
| Q105 | Only the most advanced plants have well-differentiated organs. | | | | |
| Q106 | Plant tissues have low differentiation which is the basis of vegetative propagation (asexual reproduction) | | | | |
| Q107 | Bryophytes do not have roots | | | | |
| Q108 | lack of stable character at the scale of the species gives you trouble | | | | |
| Q109 | A more evolved species does not mean that all its taxonomic characters are the most evolved | | | | |
| Q110 | In a natural environment, a species of medical interest is more important than the others. | | | | |
| Q111 | Do you regularly visit websites and forums that are interested in plants? | | | | |
| Q112 | Do you regularly visit websites that are interested in Ecology | | | | |
| Q113 | Are you interested in collecting plants (herbarium)? | | | | |
| Q114 | Teaching taxonomy is useless as plant species disappear more and more | | | | |
| Q115 | Recognizing plant species serves only to know their food and medical use as well as to reduce hunger in the world | | | | |
| Q116 | Recognizing plant species is mainly used to make herbaria and enrich gardens | | | | |
| Q117 | More DNA of individuals present similar sequences, plus the relationship between these individuals is strong | | | | |
| Q118 | Recognizing plant species serves only to know their ecological importance | | | | |

| Indicate your agreement with each of the following proposals | | Absolutely Agree | Agree | Little agree | Disagree |
|---|---|---|---|---|---|
| Q119 | The taxonomy must be studied only by naturalists | | | | |
| Q120 | Taxonomy must be studied only by ecologists | | | | |
| Q121 | Taxonomy should be taught only for pharmacists | | | | |
| Q122 | Taxonomy should be taught only for biology teachers | | | | |
| Q123 | Taxonomy must have a significant share in the media and newspapers | | | | |
| Q124 | In flowering plants, the characteristics of pollen grains are not taken into account in the current plant classification | | | | |
| Q125 | In bryophytes (mosses), the absence of conducting vessels does not allow them to reach high heights | | | | |
| Q126 | Green algae, red algae and brown algae share one common ancestor | | | | |

Q 127. Classify the following biological disciplines from the most difficult to the easiest (just use the corresponding numbers)

| 1. Biochemistry | 2. Plant Biology | 3. Genetics | 4. Plant Ecology |
|---|---|---|---|
| 5. Animal Systematic (faunistic) | 6. Animal Physiology | 7. cellular Biology | 8. Immunology |
| 9. populations Genetics | 10. Plant classification | 11. Microbiology | 12. Plant Physiology |
| most difficult | | | the easiest |

| Q 128- What disciplines of Biology do you consider essential to understand the classification of plants? | |
|---|---|
| -<br>- | -<br>- |
| -<br>- | -<br>- |

| Q129- What are the concepts relating to plant classification that pose learning difficulties? | |
|---|---|
| - | - |
| - | - |
| - | - |
| - | - |

| Q 130- Overall are you satisfied with the studies you do at university? | | | | |
|---|---|---|---|---|
| | Very satisfied | Moderately satisfied | Very little satisfied | Not satisfied |
| | | | | |

Q 131- You can add comments

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
