# Peer review of "Plant Classification Knowledge and Misconceptions among University Students in Morocco"

_education, doi:10.3390/educsci9010048_

Round 1

Reviewer 1 Report

This study aimed to identify student misconceptions and barriers to learning about plant taxonomy in a group of students from several Moroccan universities. Knowledge of plant taxonomy is necessary for students to understand the significance of biodiversity loss.  An intensive survey (135 questions) was given to student covering topics such as perception of difficulty in learning plant taxonomy, satisfaction with teaching and learning of this topic, as well as specific questions to gauge content knowledge. The authors found that most students who took this questionnaire found plant taxonomy difficult to learn, and many students reported that they learned the subject through memorization. Motivation for learning was discussed as a problem in plant taxonomy education. Students had several misconceptions about plants, their classification, and the utility of taxonomy/classification.

Broad Comments

The first paragraph is very well written. The rationale for the study and background information gave the beginning of this paper a strong start and left the reader wanting to continue to read the study.  The introduction was generally well written and although many statements were not supported by citations, a solid reasoning for the study was established.  Inclusion of more references will further strengthen the introduction.

No research or study questions were given and no hypotheses were included, so the specific aim(s) of the study were not clear. Although it is somewhat implied by the content of the introduction. 

The sample population needs further description. For example, what previous course work did students have in this field and in science in general? How are "students with a BCS degree" different from those studying for a master's degree? Presumably those studying for a master's degree have a bachelors degree. Perhaps my lack of understanding is because I don't  understand the educational system in Morocco, however, with an international readership, the educational level(s) of the study population should be clear to all readers. What were the levels of academic achievement of these students? Were they all top students? Was it a mix of high achievers, average achievers, and low achievers? What were the proportions of each?  More information on the background of these students would greatly enhance the study.

Was the survey given to students in only one particular class, such as general botany? Or in many different classes, such as general biology, general botany, and plant systematics? 

I assume the authors developed the survey. Why did the authors develop the survey in the way they did? What were the scales used for the questions? Description of survey design is needed. There is no mention of survey validity; how do the authors know that the survey is measuring what they set out for it to measure? Also no mention of survey reliability; does the survey provide consistent and dependable results each time it is used with a similar population, in similar settings? Was the survey pilot tested, or had it been used previously? Providing all of these details would greatly increase the strength of results and conclusions drawn from results.

There is no mention of any statistical tests used to estimate the significance of differences among responses.

The presentation of data and results would be greatly improved by using graphs instead of tables so that differences are visualized. The results do not give any information on the significance of differences. The way the results are presented with many tables interspersed with text made it difficult to read (choppy).

The discussion was not very compelling in the same way the introduction was. Genetics of plants was not addressed in the survey, nor in the discussion until the very end.

Questions in the survey have different numbers of possible responses; the categories of response should be consistent across all questions within a section. For example, Q3 has four options, and Q4 has three options. A standard scale should be used throughout (Likert scale is the universally accepted scale).

Some of the questions in the survey are written in such a way that they could be misleading. For example, Q95 and Q114. Also, the questions evaluating students' content knowledge should be "agree" or "disagree". There are not varying degrees of agreement or disagreement when it comes to content knowledge.

Specific Comments

Line 25: Is "Education for Sustainable Development" a special or specific program?

Lines 26 - 27: say "numerous studies" but only one citation is given.

Lines 43 - 44: citations necessary

Lines 79 - 83: unnecessary

Lines 91 - 93: can this be reframed in terms of survey response rate?

Line 94: the questionnaire had 135 questions, however, the actual survey given in the appendix is missing questions (gaps in question numbers), and the last question given is numbered Q126. Why is this?

Line 98: does "marks" mean grades?

Line 144: if this is based on Q53, 70% of students found the subject uninteresting; given the data it doesn't make sense that "almost two out of three students find this interesting"

Line 165: In Table 3, Q99, what are "higher plants" and "small plants"; is this in relation to the size of plants?

Line 179: what are "programmed notions"?

Lines 185-186: Is this being implied by questions students left blank? Can't imply they could not answer the question if they did not answer it. Could be another reason they left it blank; survey fatigue, unintentionally skipped.

Line 234: what is "tidying up"?

Lines 243 - 245: "It is important ... to increase students' motivation to learn techniques of identification of plants that have no practical use ... " The discussion would be drastically strengthened if the authors could suggest some techniques.

Lines 245 - 246: "higher plant species" might be referring to taxonomy, but then the sentence finishes with "large sizes". The meaning of this needs to be clarified.

Line 249: Lower plants or smaller plants? Why is plant size relevant? It's not clear why.

Lines 264-265: Needs further explanation. What does "confronted by other studies" mean?

Lines 268 - 269: Not had a positive impact on what?

Lines 334 - 335: For the "others" option, did students write in answers? If yes, what were they?

Line 340: Where and what are questions Q7 - Q53? This is a large omission. Even if not reported on, the complete survey should be provided in the appendix. Or reason for its omission should be explicitly stated.

Author Response

Comments and Suggestions for Authors

Author's   Reply to the Review Report (Reviewer 1)

This study aimed to identify student   misconceptions and barriers to learning about plant taxonomy in a group of   students from several Moroccan universities. Knowledge of plant taxonomy is   necessary for students to understand the significance of biodiversity loss.   An intensive survey (135 questions) was given to student covering topics such   as perception of difficulty in learning plant taxonomy, satisfaction with   teaching and learning of this topic, as well as specific questions to gauge   content knowledge. The authors found that most students who took this   questionnaire found plant taxonomy difficult to learn, and many students   reported that they learned the subject through memorization. Motivation for   learning was discussed as a problem in plant taxonomy education. Students had   several misconceptions about plants, their classification, and the utility of   taxonomy/classification.

Broad Comments

The first paragraph is very well written.   The rationale for the study and background information gave the beginning of   this paper a strong start and left the reader wanting to continue to read the   study. The introduction was generally well written and although many   statements were not supported by citations, a solid reasoning for the study   was established. Inclusion of more references will further strengthen the   introduction.

More citations were aded

No research or study questions were given   and no hypotheses were included, so the specific aim(s) of the study were not   clear. Although it is somewhat implied by the content of the introduction.

Done 

The sample population needs further   description. For example, what previous course work did students have in this   field and in science in general? How are "students with a BCS   degree" different from those studying for a master's degree? Presumably   those studying for a master's degree have a bachelors degree. Perhaps my lack   of understanding is because I don't understand the educational system in   Morocco, however, with an international readership, the educational level(s)   of the study population should be clear to all readers. What were the levels   of academic achievement of these students? Were they all top students? Was it   a mix of high achievers, average achievers, and low achievers? What were the   proportions of each? More information on the background of these students   would greatly enhance the study.

Done

Methodology was detailed and the various   informations requested have been incorporated into the text (see Methods   & materials)

Was the survey given to students in only one   particular class, such as general botany? Or in many different classes, such as   general biology, general botany, and plant systematics?

the survey was given to students in many   different classes

- I assume the authors developed the survey.   Why did the authors develop the survey in the way they did? -   

- What were the scales used for the   questions?

- Description of survey design is needed. There is no   mention of survey   validity; how do the authors know that the survey is measuring what   they set out for it to measure? Also no mention of survey reliability; does the survey   provide consistent and dependable results each time it is used with a similar   population, in similar settings? Was the survey pilot tested, or had it been used   previously? Providing all of these details would greatly increase the   strength of results and conclusions drawn from results.

- Methodology has   been developed in more detail

- We have   used Likert scale (see Questionnaire in Appendix)

Done

There is no mention of any statistical tests   used to estimate the significance of differences among responses.

The presentation of data and results would   be greatly improved by using   graphs instead of tables so that differences are visualized. The results do not give any   information on the significance of differences. The way the results   are presented with many tables interspersed with text made it difficult to   read (choppy).

Done (tables were replaced by graphs)

The discussion was not very compelling in   the same way the introduction was. Genetics of plants was not addressed in   the survey, nor in the discussion until the very end.

Discussion was improved by adding relevant   elements to analysis of the results

- Questions in the survey have different   numbers of possible responses; the categories of response should be   consistent across all questions within a section.

- For example, Q3 has four options, and Q4   has three options. A standard scale should be used throughout (Likert scale   is the universally accepted scale).

- we adjust the number of options for each   type or group of questions

- Question Q4 has four options

(very difficult  -   difficult- moderately difficult-  easy)   

- Some of the questions in the survey are   written in such a way that they could be misleading. For example, Q95 and Q114.

- Also, the questions evaluating students'   content knowledge should be "agree" or "disagree". There   are not varying degrees of agreement or disagreement when it comes to content   knowledge.

- For Q114 we want to know if the student   understands the importance of classification.

it is because of the extinction of   thousands of species that scientists urgently need to list all species before   they are extinguished

- Correct : both boxes (absolutely Agree   and Agree) have been grouped, as well for the other two. Incorrect answers   have been identified and accounted for (this is what is presented in the   results)

Specific Comments

Line 25: Is "Education for Sustainable   Development" a special or specific program?

"Education   for Sustainable Development"  as a   separate subject or integrated into other disciplines (the most frequent   case)

Lines 26 - 27: say   "numerous studies" but only one citation is given.

More   citations are aded

Lines 43 - 44: citations necessary

done

Lines 79 - 83:   unnecessary

removed

Lines 91 - 93: can this   be reframed in terms of survey response rate?

done

Line 94: the questionnaire had 135   questions, however, the actual survey given in the appendix is missing   questions (gaps in question numbers), and the last question given is numbered   Q126. Why is this?

the complete questionnaire is provided   in the appendix

Line 98: does   "marks" mean grades?

Yes (the   American word is grade)

Line 144: if this is based on Q53, 70% of   students found the subject uninteresting; given the data it doesn't make sense   that "almost two out of three students find this interesting"

70% of   students  disagree with the proposition “I do not find this class   interesting, so I work with the minimum of my effort” i.e  only 30% of students find this class   uninteresting

Line 165: In Table 3, Q99, what are   "higher plants" and "small plants"; is this in relation   to the size of plants?

Yes,  "higher plants" mean large plants   (eg trees in relation to shrubs)

Line 179: what are   "programmed notions"?

it means   taught concepts

Lines 185-186: Is this being implied by   questions students left blank? Can't imply they could not answer the question   if they did not answer it. Could be another reason they left it blank; survey   fatigue, unintentionally skipped.

these are   probable reasons

Line 234: what is   "tidying up"?

tidying up   plants means put plants in order

Lines 243 - 245: "It is important ...   to increase students' motivation to learn techniques of identification of   plants that have no practical use ... " The discussion would be   drastically strengthened if the authors could suggest some techniques.

Done

Lines 245 - 246: "higher plant   species" might be referring to taxonomy, but then the sentence finishes   with "large sizes". The meaning of this needs to be clarified.

It can also   be noticed that textbooks in Moroccan secondary schools often include higher   plant species; thus, students become familiar with species with large sizes.

- it means that the fact that   textbooks ofen represent “higher plants” with “large size species” (trees and   shrubs) creates confusion for students who may associate the   "higher" (taxonomic) attribute with "larger" ( adjective   of size)

Line 249: Lower plants or smaller plants?   Why is plant size relevant? It's not   clear why.

see previous comment

Lines 264-265: Needs further explanation.   What does "confronted by other studies" mean?

These results   have been found in   other studies on students' conceptions of plants

Lines 268 - 269: Not   had a positive impact on what?

Corrected  (….. a positive impact on understanding and   acceptance of evolution)

Lines 334 - 335: For the "others"   option, did students write in answers? If yes, what were they?

- the curriculum of this module is too   busy

-    time allotted for this module is insufficient

-    number of lab sessions is insufficient

- difficulty understanding the basics of   taxonomy

-    number of species to memorize is too large

- Morphological characters require great   expertise

- some identification characters appeal to   notions of plant biology that we forgot

- course of taxonomy requires memorization

-    rhythm of the lectures of the taxonomy is fast

- scarcity of group work sessions

- this module does not motivate me

Line 340: Where and what are questions Q7 -   Q53? This is a large omission. Even if not reported on, the complete survey   should be provided in the appendix. Or reason for its omission should be   explicitly stated.

the complete   questionnaire is provided in the appendix

Reviewer 2 Report

After reading “Plant classification teaching and learning in Moroccan university: Student’s knowledge and misconceptions” I feel like the latter half of the title is really more descriptive about what this paper showed.

There was very little about teaching and learning, the paper was mostly about student’s knowledge and misconceptions. I think the paper should be re-framed around this because these are the data that are presented.

Overall I feel like the paper failed to show me that a student’s knowledge of plant taxonomy is an important metric to measure. The introduction had several bold statements without references to back them up. For example, the first 3 sentences of the paper make big claims but provide no evidence (e.g., lack of awareness leads to practices overexploiting natural resources and damaging biodiversity).  Furthermore, it is unclear how a students knowledge of the facts about plant reproductive biology (something pointed out in the paper) can influence their tendency to overexploit natural resources.

Other similarly bold statements show that the researchers like plant taxonomy (e.g., line 37: “Plant taxonomy teaching is the most important part of botany in universities…”), but the paper did not clearly explain why plant taxonomy learning is so important.  Different ideas were mentioned, but not really convincingly fleshed out.

Surveys like this given to students are useful, but the results need to be interpreted very cautiously as the answers given are greatly influenced by the way the question was written. For example, question 97 about current species and fossil species being considered in a similar way. Just because students disagree doesn’t mean they don’t think fossils are used in taxonomy, it could be related to the understanding that current classifications of many taxa are increasingly based on DNA, which is not available for fossil taxa. Or question 61 “there was an evolution in the plants kingdom” 27% disagreed, but this doesn’t necessarily mean 26% of people do not believe in evolution of plants, they might simply misunderstand the question. It is difficult to know.

Also, the interpretation of the student answers needs to be done more cautiously. For example, in question 116 36% of students agreed with the statement that recognizing plant species is mainly used to make herbaria and enrich gardens, but, in the results section the authors state that “ one out of three students thinks that the sole objective of plant classification is to make herbaria and enrich gardens”. The key difference here is the question says “mainly” and the authors interpreted the results as “the sole objective”. These are different things.

Also the authors often overstate student beliefs. E.g., line 183-184, “one out of three students believe…” but the presented percentages are ~21% and ~14%, neither of those show the stated 1/3 of students.

In the discussion there is a lot of speculation that is not scientifically sound. For example line 240 “They think that recognizing plant species only serves to know their food and medical uses”. I assume this statement comes form question 115 “recognizing plant species serves only to know their food and medical use…”. But 60% of students disagreed with that statement, which seems to be the opposite of what the discussion in the paper says.

Further examples of speculation include line 270, “they believe that evolution has been primarily related to the human species.” What is this based on, I did not see this question in the survey.

Overall, while I think it is important to measure students understanding of various subjects, including plant taxonomy, this paper does not discuss the teaching and learning aspects, nor does it honestly depict the student perceptions of plant taxonomy or even the broad importance of teaching plant taxonomy to a diverse set of biology students. The survey results are, however useful if the question being asked can be more focused and speculation about the answers can be avoided. I suggest the authors go back and re-work their paper to better describe simply the lack of understanding students have about plant taxonomy.

Author Response

Comments and Suggestions for Authors

Author's Reply to the Review Report (Reviewer 2)

This manuscript describes the results of a   questionnaire administered to botany students in several universities in   Morocco. The results revealed significant misconceptions and motivational   shortcomings in the students.

As such, the paper provides a signal that   the educational approaches need to be changed to address these issues. The   manuscript has an interesting objective, but there are a number of issues   that need to be addressed before it is ready for publication.

The manuscript would benefit from some   additional data analysis. The

results are based on responses from 737   students from four universities.

What is not stated is how many classes and   how many different professors contributed to the data.

It would be informative to know that the   misconceptions and attitude issues are equally represented across all   institutions, professors, and student levels. Otherwise, the pattern of   results could be attributed to one or two particularly ineffective   instructors.

Generally, several professors participate   in the teaching of a module (1 or 2 for lectures, more than 4 for tutorials   and practical work) and this for each of the institutions surveyed.

  Therefore, it seems difficult to   associate a group of students to a teacher because this was not taken into   account when the questionnaire was submitted.

However, the curriculum is the same for   all students surveyed and we find that identified  misconceptions are present among the   students of all the faculties that took part in this survey.

Lines 116-120 are unclear without an   explanation of what “average” refers to. It appears that the most relevant   average would be the mean of the class, but perhaps there is a different   system being described here.

Corrected : the   average of an exam  refers to a mark   equal to 10/20.           this is the   minimum score to validate a module

- In some areas, the authors misrepresent   the data. For example, on line 172, it is stated that “one of three students   thinks that the sole objective of plant classification is to make   herbaria and enrich gardens.” The actual wording of question 116, however, is   that “the main objective is to make herbaria and enrich   gardens” – a far less extreme statement.

- As a second example, on line 224 the   authors state that “almost half of the students assert that plant taxonomy’s   sole objective is to determine ‘which species is close to which.’”It is   unclear from the manuscript that this is exactly how question 89 is

worded.

-In lines 266-273, the authors indicate that   “more than a quarter of students do not believe in plant evolution.” This   appears to be based on reference 24, but I do not see that evidence in that   article. (I have not checked the authors’ accuracy in their representation of   other articles.)

- Corrected :   “one of three students thinks that the main objective of plant classification is to make   herbaria and enrich gardens.”

 -Corrected : “almost half of the students   assert that plant taxonomy’s objective is to determine ‘which species is   close to which.’”

- This refers   to the answers to question Q61 (26.74% of students surveyed disagree with   fact that there was an evolution in plants kingdom). but this result was   already confirmed by other research   [45,46,47]

N°of reference   was corrected [45,46,47]

Although the manuscript makes the case that   instruction needs to be

changed in order to deal with the   shortcomings in students’ understanding and motivation, there are relatively   few suggestions of exactly how they recommend that this be accomplished.

Some suggestions are added

- Finally, although the manuscript shows the   authors’ impressive command of the English language, there are many places   where sentences have grammatical errors and problems in expression. Perhaps   from problems in translation, some of the items of the questionnaire (e.g.,   72, 89, 90, 94, 97,) are not well expressed. If the wording of these items   was accurate, then it presents a difficulty in interpreting student responses   for those items.

- Regardless, it would be helpful for the authors   to seek the assistance of a native speaker of English to polish up the   writing.

- The   questionnaire was written in French, the English translation probably led to   changes of meaning.

These   questions have been reformulated

- The article   will be reviewed in relation to the language to improve its understanding

Reviewer 3 Report

This manuscript describes the results of a questionnaire administered to botany students in several universities in Morocco.  The results revealed significant misconceptions and motivational shortcomings in the students.  As such, the paper provides a signal that the educational approaches need to be changed to address these issues.  The manuscript has an interesting objective, but there are a number of issues that need to be addressed before it is ready for publication.

The manuscript would benefit from some additional data analysis.  The results are based on responses from 737 students from four universities.  What is not stated is how many classes and how many different professors contributed to the data.  It would be informative to know that the misconceptions and attitude issues are equally represented across all institutions, professors, and student levels.   Otherwise, the pattern of results could be attributed to one or two particularly ineffective instructors.

Lines 116-120 are unclear without an explanation of what “average” refers to. It appears that the most relevant average would be the mean of the class, but perhaps there is a different system being described here.

In some areas, the authors misrepresent the data.  For example, on line 172, it is stated that “one of three students thinks that the sole objective of plant classification is to make herbaria and enrich gardens.”  The actual wording of question 116, however, is that  “the main objective is to make herbaria and enrich gardens” – a far less extreme statement.  As a second example, on line 224 the authors state that “almost half of the students assert that plant taxonomy’s sole objective is to determine ‘which species is close to which.’” It is unclear from the manuscript that this is exactly how question 89 is worded.  In lines 266-273, the authors indicate that “more than a quarter of students do not believe in plant evolution.” This appears to be based on reference 24, but I do not see that evidence in that article. (I have not checked the authors’ accuracy in their representation of other articles.)

Although the manuscript makes the case that instruction needs to be changed in order to deal with the shortcomings in students’ understanding and motivation, there are relatively few suggestions of exactly how they recommend that this be accomplished.

Finally, although the manuscript shows the authors’ impressive command of the English language, there are many places where sentences have grammatical errors and problems in expression.  Perhaps from problems in translation, some of the items of the questionnaire (e.g.,  72, 89, 90, 94, 97,  ) are not well expressed.  If the wording of these items was accurate, then it presents a difficulty in interpreting student responses for those items.  Regardless, it would be helpful for the authors to seek the assistance of a native speaker of English to polish up the writing.

I hope that the authors find these comments helpful.

Author Response

Comments and   Suggestions for Authors

        Author's Reply to the Review Report   (Reviewer 3)

After reading   “Plant classification teaching and learning in Moroccan university: Student’s   knowledge and misconceptions” I feel like the latter half of the title is   really more descriptive about what this paper showed

There was very   little about teaching and learning, the paper was mostly about student’s   knowledge and misconceptions. I think the paper should be re-framed around   this because these are the data that are presented. 

I think that the   title should be changed to reflect the content of the article.

We propose the   following title:

 “Plant   classification Knowledge and misconceptions among   university Students in Morocco

Overall I feel   like the paper failed to show me that a student’s knowledge of plant taxonomy   is an important metric to measure. The introduction had several bold   statements without references to back them up. For example, the first 3   sentences of the paper make big claims but provide no evidence (e.g., lack of   awareness leads to practices overexploiting natural resources and damaging   biodiversity).  Furthermore, it is unclear how a students knowledge of   the facts about plant reproductive biology (something pointed out in the   paper) can influence their tendency to overexploit natural resources.

- more  references were added

- there is a   relationship between the scientific knowledge and citizen behavior and   responsibility. However, this link between knowledge and social awareness   must be clearly prescribed. the behavior expected from students with regard   to biodiversity education must be more prescribed by school.

Other similarly   bold statements show that the researchers like plant taxonomy (e.g., line 37:   “Plant taxonomy teaching is the most important part of botany in   universities…”), but the paper did not clearly explain why plant taxonomy   learning is so important.  Different ideas were mentioned, but not   really convincingly fleshed out. 

I mean that plant   taxonomy is an   important part of Plant Biology program at the University

- Surveys like   this given to students are useful, but the results need to be interpreted   very cautiously as the answers given are greatly influenced by the way the   question was written. For example, question 97 about current species and   fossil species being considered in a similar way. Just because students   disagree doesn’t mean they don’t think fossils are used in taxonomy, it could   be related to the understanding that current classifications of many taxa are   increasingly based on DNA, which is not available for fossil taxa.

- Or question 61   “there was an evolution in the plants kingdom” 27% disagreed, but this   doesn’t necessarily mean 26% of people do not believe in evolution of plants,   they might simply misunderstand the question.  It is difficult to know

- I agree with you   about question 97, it may be a knowledge gap on this topic

- For question 61,   there is an influence of beliefs on everything related to biological   evolution. This was noted in interviews (the results of which are not   mentioned in this article) as it has already been shown by other research   whose references have been added to the text ([45,46,47])

but in general, we   must always nuance results related    questions about attitudes and perceptions

Also, the   interpretation of the student answers needs to be done more cautiously. For   example, in question 116 36% of students agreed with the statement that   recognizing plant species is mainly used to make herbaria and enrich gardens,   but, in the results section the authors state that “ one out of three   students thinks that the sole objective of plant classification is to make   herbaria and enrich gardens”. The key difference here is the question says   “mainly” and the authors interpreted the results as “the sole objective”.   These are different things

Corrected : “one of three students thinks   that the main   objective of plant classification is to make herbaria and enrich gardens.”

Also the authors   often overstate student beliefs. E.g., line 183-184, “one out of three   students believe…” but the presented percentages are ~21% and ~14%, neither   of those show the stated 1/3 of students. 

Corected

In the   discussion there is a lot of speculation that is not scientifically sound.   For example line 240 “They think that recognizing plant species only serves   to know their food and medical uses”. I assume this statement comes form   question 115 “recognizing plant species serves only to know their food and   medical use…”. But 60% of students disagreed with that statement, which seems   to be the opposite of what the discussion in the paper says

They……”      was substituted  by “some of students surveyd   think that recognizing plant……( 40%). 

Further examples   of speculation include line 270, “they believe that evolution has been   primarily related to the human species.” What is this based on, I did not see   this question in the survey. 

This refers to the answers to question Q61   (26.74% of students surveyed disagree with fact that there was an evolution   in plants kingdom)

the fact that   Moroccan students do not believe in biological evolution has been found in   other research ([45,46,47])

Overall, while I   think it is important to measure students understanding of various subjects,   including plant taxonomy, this paper does not discuss the teaching and   learning aspects, nor does it honestly depict the student perceptions of   plant taxonomy or even the broad importance of teaching plant taxonomy to a   diverse set of biology students. The survey results are, however useful if   the question being asked can be more focused and speculation about the   answers can be avoided. I suggest the authors go back and re-work their paper   to better describe simply the lack of understanding students have about plant   taxonomy

Title, research   questions and the text have been reworded to better describe the results. we   hope to have satisfied your relevant remarks

Round 2

Reviewer 1 Report

Thank you for addressing most of my comments and making suggested edits.  This version is much improved.  Upon second review, I have additional questions and suggestions. 

Broad Comments

 The abstract does not accurately summarize the manuscript; semi-structured interviews are not included in the methods section.  The abstract can be improved to better reflect the study, it’s relevance, and outcomes.

The additional references give more credibility to the statements of the introduction; I have not gone through them each specifically and trust the authors know the literature pertaining to this topic.  Citations in additional places would further strengthen the introduction (please see specific comments below).

What changes in plant taxonomy/classification education have been implemented?  Have the impacts of these changes been measured?

The research questions are adequate.

Section 2.1 should be edited so that it is succinct and focused on relative information only. I found this confusing to read.  I found lines 117 – 118 to be clear and relevant.

What do the practical work and tutorials consist of? Is the practical work lab exercises?

Both undergraduate and master's degree students were surveyed, but their responses are lumped together.  However, presumably master's degree students have taken more courses and therefore have more education in relation to plant classification than undergraduates do. This could affect results and is not discussed in the manuscript.

Additionally, were the students who responded to the survey mostly high achievers overall or was there a range of academic abilities represented in this sample? Academic preparation and ability could affect these survey results.

Was there an informed consent process for students taking the survey?  Was institutional permission via an Institutional Review Board (IRB) required for use of human subjects?  This information should be included in the methods.

Was the survey paper-based or completed electronically?

I am not sure that I understand the educational levels of survey respondents (lines 132 – 133).  For Baccalaureate degree + two years, are these students who already earned their BSC and have gone on to study for an additional two years? If yes, how are these students different than the master's degree students? Are the BSC degree students those who are studying towards their baccalaureate degree?

Please label x-axis in Figures 2, 3, 4, 5, 6, and 7.

Are survey questions Q118 and Q115 in Figure 4 mutually exclusive? Or could a student agree that recognizing plant species serves both to know ecological importance and food/medical use?

Overall, the manuscript would benefit by being further editing for clarity of English language use.

Specific Comments

Lines 23 – 24: “sustainable exploitation” is a bit of an oxymoron; I suggest using a different word than exploitation, like “use” instead.

Line 24: If “Education for Sustainable Development” is not a specific program than it should not be in caps.

Line 25 - 26: It would be helpful to give a little more detail as to how the “numerous studies” have shown education and training to be “the most effective lever for preserving biodiversity”.  This is a strong and broad statement that would be more credible with further explanation.

Lines 36-37: Can a citation for “… is the focus of all the reforms of this branch of education” be provided?

Lines 37-38: What are “…the existing problems in plant taxonomy teaching”? What are some of the “reforming measures” that have been suggested? The authors provide the focus of reforms, but not what the suggested reforms actually are.

Lines 38-40: This sentence would benefit from having citations to back-up the statement.

Lines 43-44: Sentence needs a citation (“Research has shown …”).

Line 45: Please use “humans” or “people” instead of “man”; women also can plant flowers.

Line 74: Consider revising as “ … an approach to studying these problems.”  The preceding paragraph does not give any “problematic questions”.

Line 120: “In order to arrive at the desired result …” please edit for meaning. If research is to be as objective as possible, then researchers should not be trying to arrive at a desired result. 

Line 147: Are the experts referred to content knowledge experts or other type?

Lines 157 – 158: How does the survey having four parts, “…avoid any possible ambiguity”?

Lines 161 – 164: Are these the four parts referred to above?

Line 167: Please delete “(Questions Q1 and Q2)”; the question numbers are not given in any other subheadings.

Lines 168-171: Why are students graded on 20 points? Is this a final exam? This result is difficult to understand without knowledge of how students are graded. Are they graded on 20 exam questions?

Lines 215 – 216: There is no Table 3; do the authors mean Figure 4?

Lines 257-258: What is “quite a large number”? Based on the results in this paragraph, it seems that more students answered questions correctly than incorrectly.  Although I see in Figure 8 there are two questions which more than 50% answered incorrectly.

Lines 267 – 268: But six out of ten students could answer questions related to concepts they had already studied; is the four out of ten students a significant number of students?

Lines 377 – 381: Please consider moving this to the introduction, or at least also give this information in the introduction. This is very relevant to the context of the study and should be upfront.

Author Response

Reviewer 1

Title   : Plant Classification   Knowledge and Misconceptions among University Students in Morocco

Comments and Suggestions for Authors

Author's Reply to the Review Report (Reviewer 1)

Broad   Comments

 The abstract does not accurately summarize   the manuscript; semi-structured interviews are not included in the methods   section.

The   abstract can be improved to better reflect the study, it’s relevance, and   outcomes.

The   additional references give more credibility to the statements of the   introduction; I have not gone through them each specifically and trust the   authors know the literature pertaining to this topic.  Citations in additional places would   further strengthen the introduction (please see specific comments below).

 What changes in plant   taxonomy/classification education have been implemented?  Have the impacts of these changes been   measured?

The   research questions are adequate.

 The semi-structured   interview was conducted but results are not included in the article   (reference to that in abstract was deleted)

some   results were incorporated into summary

more   citations have been added

In   Morocco, the changes concern rather the increase of the allocated hourly   volume to teaching plant systematics

There   is no study on the evaluation of the teaching/learning of plant   classification, the present study aims in part to evaluate it

Section 2.1 should be edited so that it is succinct and focused on   relative information only. I found this confusing to read.  I found   lines 117 – 118 to be clear and relevant.

Done

What do the practical work and tutorials consist of? Is the practical   work lab exercises?

Tutorial   works aim to deepen or complete the concepts studied in the lecture

During   practical work ( lab   class), students make manipulations (eg make observations under microscope,   dissect flowers ...), this is the practical part of the course

 Both undergraduate and master's degree students were surveyed, but   their responses are lumped together.  However, presumably master's   degree students have taken more courses and therefore have more education in   relation to plant classification than undergraduates do. This could affect results   and is not discussed in the manuscript.

we   did not want to analyze the impact of the level of studies on learning,   otherwise we would have separated student responses.But since Master's   students make up only 26% of our sample, we opted not to consider this variable   in this article

Additionally, were the students who responded to the survey mostly high   achievers overall or was there a range of academic abilities represented in   this sample? Academic preparation and ability could affect these survey   results.

The   sample was not selected on the basis of the academic abilities of students,   so it is made up of students who have different levels and academic abilities

Was there an   informed consent process for students taking the survey?  Was   institutional permission via an Institutional Review Board (IRB) required for   use of human subjects?  This information should be included in the   methods

Ethical approval and Data   Collection

 More details on ethical were included in the methodology (see p5 )

 Was the survey paper-based or completed electronically?

The   survey was paper-based

This   information was included in methodology

I am not sure   that I understand the educational levels of survey respondents (lines 132 –   133).  For Baccalaureate degree + two years, are these students who   already earned their BSC and have gone on to study for an additional two   years? If yes, how are these students different than the master's degree   students? Are the BSC degree students those who are studying towards their   baccalaureate degree?

Baccalaureate degree mean High School Diploma

Baccalaureate degree + two years = High School Diploma + 2 years (associate Degree)

Baccalaureate degree + 3 years = High School Diploma + 3 years = Bachelor of science

 - Bachelor of science + 1 year = Bachelor of   Honor = Master 1 = High School   Diploma +4 years

master's degree   = Bachelor of science + 2 years =   Master of science

This   information was included in methodology

Please label x-axis in Figures 2, 3, 4, 5, 6, and 7.

x-axis shows the  prcentage of  responses

Done

 Are survey questions Q118 and Q115 in Figure 4 mutually exclusive?   Or could a student agree that recognizing plant species serves both to know   ecological importance and food/medical use?

No,   the two questions are not mutually exclusive; a student can agree that recognizing plant species serves both to know   ecological importance and food/medical use

Specific Comments

Lines 23 – 24: “sustainable exploitation” is a bit of an oxymoron; I suggest   using a different word than exploitation, like “use” instead.

Done  

Line 24: If “Education for Sustainable Development” is not a specific   program than it should not be in caps.

Done    (education for   sustainable development)

Line 25 - 26: It would be helpful to give a little more detail as to how   the “numerous studies” have shown education and training to be “the most   effective lever for preserving biodiversity”.  This is a strong and   broad statement that would be more credible with further explanation.

clarifications were made to   this part of the article (see page 1)

Lines 36-37: Can a citation for “… is the focus of all the reforms of   this branch of education” be provided?

Done

Lines 37-38: What are “…the existing problems in plant taxonomy teaching”?   What are some of the “reforming measures” that have been suggested? The   authors provide the focus of reforms, but not what the suggested reforms   actually are.

In   order to promote plant taxonomy teaching

these measures are indicated in p14 lines 364-373).

Lines 38-40: This sentence would benefit from having citations to back-up   the statement.

Citation   are added

Lines 43-44: Sentence needs a citation (“Research has shown …”).

Done

Line 45: Please use “humans” or “people” instead of “man”; women also can   plant flowers.

Done

Line 74: Consider revising as “ … an approach to studying these   problems.”  The preceding paragraph does not give any “problematic   questions”.

corrected   (the sentence “all in all…..” was deleted)

Line 120: “In order to arrive at the desired result …” please edit for   meaning. If research is to be as objective as possible, then researchers   should not be trying to arrive at a desired result. 

Corrected   : “  In order to study knowledge   and  identify misconceptions….”

Line 147: Are the experts referred to content knowledge experts or other   type?

the   terme “experts” referred both to content knowledge experts and french   language ones

(added   in text)

Lines 157 – 158: How does the survey having four parts, “…avoid any   possible ambiguity”?

survey   questions can be classified into four sections

Lines 161 – 164: Are these the four parts referred to above?

yes

Line 167: Please delete “(Questions Q1 and Q2)”; the question numbers are   not given in any other subheadings.

Done

Lines 168-171:

Why are students graded on 20 points? Is this a final exam?

This result is difficult to understand without knowledge of how students   are graded.

Are they graded on 20 exam questions?

No,  it's the average of exam in the middle of   the semester and the final exam

whatever   the number of questions asked, professor always brings back the maximum score   that a student can obtain at 20/20

below   the correspondence between the Moroccan and English school marks system

18/20   - 20/20 -->A+ 
  15 -18 -->A
  14 - 15-->A- 
  12-13 -->B+
  10-11 -->B 
  9 ------>B-

Lines 215 – 216: There is no Table 3; do the authors mean Figure 4?

Sorry,   we mean Figure 4

(corrected   in text)

Lines 257-258: What is “quite a large number”? Based on the results in   this paragraph, it seems that more students answered questions correctly than   incorrectly.  Although I see in Figure 8 there are two questions which   more than 50% answered incorrectly.

“quite a large   number” replaced by “a part of students”

Lines 267 – 268: But six out of ten students could answer questions   related to concepts they had already studied; is the four out of ten students   a significant number of students?

yes,   that's about half of the surveyed students. it seems that the students have not assimilated the taught course

Lines 377 – 381: Please consider moving this to the introduction, or at   least also give this information in the introduction. This is very relevant   to the context of the study and should be upfront.

Done

Submission Date  11 January 2019

Date of this review   19 Feb 2019 05:25:12

Reviewer 2 Report

I am satisfied with the changes the authors made to the manuscript

Author Response

Reviewer 2

Journal    Education Sciences (ISSN 2227-7102)

Manuscript ID education-434928

Type Article

Number of Pages 13

Title : Plant classification teaching   and learning in Moroccan university: Student’s knowledge and misconceptions

Abstract

This study aims at assessing learning   outcomes and identifying students’ misconceptions in plant classification.   Hence, we conducted a questionnaire survey and semi-structured interviews   with undergraduate and master’s students. The qualitative analysis of the   students' responses made it possible to shed light on difficulties of   assimilation of many notions and also to identify the different   misconceptions constructed during their learning courses about plant   organisms. Thanks to it, we could see different types of problems in plant classification,   which constitute misconceptions hindering learning. Initial training in   plant’s biology does not appear to have a significant effect in modifying   students' misconceptions related to plant classification.

Comments and Suggestions for Authors : I am   satisfied with the changes the authors made to the manuscript

Submission Date          11 January 2019           Date of this review        12 Feb 2019 05:43:40

Thanks for relevant comments and suggestions

Reviewer 3 Report

This version of the manuscript is improved in focus, detail, and clarity, but there are still shortcomings.  The general issues fall into three categories:

1.     Errors in the use of English.  In my previous review, I recommended that the authors have someone who is a native English speaker examine the manuscript.  This has clearly not been done.  If the current version is accepted for publication, then the journal editors should have someone copyedit the text and make corrections.  For examples of the errors, see lines 71-73, 168-169, 260-261, 283-284, 310, and 377.

2.     Accuracy of citations.  I examined Reference 23 regarding lines 46 and 47.  I saw nothing in that reference that related to the statement in the manuscript.  I did not examine the other references for their accuracy, but the authors should check all references.

3.     Lack of clarity in the questionnaire.  This might be related to issues of translation, but there are still questions that do not make sense or are ambiguous (e.g., Q 70, Q 89).

Some specific issues:

Lines 85-86         It is unclear why the designations TD and TP are included.  They are rarely used in the manuscript and do not have a logical relationship with the words that they stand for.

Line 92     discipline

Lines 168-171     Besides the grammatical errors, the indication that 1/3 of the respondents have lower than average grades can indicate 1) that the sample was affected because the rest of the below average students did not respond, 2) that the distribution of grades was skewed, or 3) that the term “average” was not used to indicate the mean score of the group but was an arbitrary term that really meant “satisfactory.”  I assume that that last of these was the case, and if so, then the term “satisfactory” would be better to use than “average.”

Line 175   The heading of this section refers to difficulty in learning, but this paragraph talks about difficulty in teaching. Which is it?

Lines 179-180  What questions give rise to these results?  Less important than what?

Lines 179-183  The juxtaposition of statements here make it impossible to know what is being communicated.  For example, one might easily state that something is important but less important than something else.  What do we take from that?

Line 210 In a similar way to what? The question was unclear, so it is impossible to understand what to take from the results.

Line 215  more motivated than what?

Line 217 than other what?

Lines 246-247     It is not meaningful to the reader to just see the question numbers.

Lines 267-268     Is this meant to contrast with items that the students had not studied?  If most of the students are performing above average (whatever that means) then how is this to be interpreted?

I hope that the authors find these observations helpful.

Author Response

Manuscript ID: education-434928
  Title: Plant classification teaching and   learning in Moroccan university: Student’s knowledge and misconceptions

Revisions in the manuscript and responses to the reviewers'   comments.

At first I want to thank reviewers for their relevant comments  and suggestions which will help improve this   article

Reviewer 3

The reviewer comments   are relevant and the suggestions helped us to improve the quality of the   article, thank you

Comments and Suggestions for Authors

This version of the manuscript is improved in focus,   detail, and clarity, but there are still shortcomings.  The general   issues fall into three categories: 

Comments and Suggestions for Authors

Author's Reply to the Review Report (Reviewer 3)

1.     Errors in the use of English.  In my previous   review, I recommended that the authors have someone who is a native English   speaker examine the manuscript.  This has clearly not been done.    If the current version is accepted for publication, then the journal editors   should have someone copyedit the text and make corrections.  For   examples of the errors, see lines 71-73, 168-169, 260-261, 283-284, 310, and   377.

Before its publication the   text will be

copyedit  and the necessary corrections will

be made.

2.     Accuracy of citations.  I examined Reference 23   regarding lines 46 and 47.  I saw nothing in that reference that related   to the statement in the manuscript.  I did not examine the other   references for their accuracy, but the authors should check all references.

All references were checked

3.     Lack of clarity in the questionnaire.  This might   be related to issues of translation, but there are still questions that do   not make sense or are ambiguous (e.g., Q 70, Q 89).

questions have been reformulated as follow

Q 70 :   I learn better when the   studied plant is useful

Q 89 : the taxonomy of plants is to determine   "which plant is close to other"

Some specific issues:

Lines   85-86         It is unclear why the   designations TD and TP are included.  They are rarely used in the   manuscript and do not have a logical relationship with the words that they   stand for.

“TD and TP” are replaced by

tutorial works  and practical works

Line 92     discipline

“Disciplne” means “specialty”

this paragraph has been changed   and the word "discipline" has been deleted

 Lines 168-171     Besides the   grammatical errors, the indication that 1/3 of the respondents have lower   than average grades can indicate 1) that the sample was affected because the   rest of the below average students did not respond, 2) that the distribution   of grades was skewed, or 3) that the term “average” was not used to indicate   the mean score of the group but was an arbitrary term that really meant   “satisfactory.”  I assume that last of these was the case, and if so,   then the term “satisfactory” would be better to use than “average.”

the term “average” meant “satisfactory.” 

The the term “average” was replaced by “satisfactory.”

term “average” mean that student had a mark = 10/20 wich correpond to “B” in   English school marks system

Line 175   The heading of this section refers   to difficulty in learning, but this paragraph talks about difficulty in   teaching. Which is it?

paragraph concerns the   perceptions of students about learning difficulties . The term “teaching” was   replaced by “learning” (line 220)

Lines 179-180  What questions give rise to these   results?  Less important than what?

It’s the question Q3 (What do you think about the   importance of plant systematics ?)

the sentence   has been corrected

Three out of four students said   that plant systematics is important (74.2%) while one out of four considered   it to be a litte or not  important   disciplin

 Lines 179-183  The juxtaposition of statements here make it   impossible to know what is being communicated.  For example, one might   easily state that something is important but less important than something   else.  What do we take from that?

this may indicate that students do not seem to pay enough interest in   learning plant classification because they do not think that this specialty   is important for other biological specialties.

Line 210 In a similar way to what?

The question was unclear, so it is impossible to understand what to take   from the results

The ranking difficulties are accentuated when it comes to fossil plants.

Corrected

It mean that students seem to   think that current species are more important in taxonomy than extinct or   fossil species because of the characters needed for identification and   classification (morphological, anatomical, biochemical or genetic characters)   are easier to use in current species than in fossil (especially for DNA-based   identification).

              Corrected in text

. Line 215  more motivated than what?

Corrected

we notice that   the majority of students are more   motivated

Line 217 than other what?

Than other species without   medical use

Lines 246-247     It is not   meaningful to the reader to just see the question numbers.

questions were provide

Lines 267-268       Is this meant to contrast with items that the students had not   studied?  If most of the students are   performing above average (whatever that means) then how is this to be   interpreted?

Almost 4 out of 10 students   (39.9%) could not correctly answer the questions related to the notions   already studied.

Having a score above   "satisfactory" does not mean that the student had understood all   the concepts programmed in the course of plant biology, questions 87, 96 and   106 illustrate this. in other words, the notions relating to plant   reproduction (the place of the gametophyte phase and sporophute in the life   cycle of angiosperms, autogamy reduces diversity in plants) or to histology   are poorly understood by the students.

I hope that the authors find these observations   helpful.

the comments are relevant   and the suggestions helped us to improve the quality of the article, we thank   you

Submission Date : 11 January 2019     Date of this review  18 Feb 2019 23:45:42

Journal  Education Sciences (ISSN 2227-7102)      Manuscript   ID  education-434928
